# Scene Graph Enhanced Pseudo-Labeling for Referring Expression Comprehension

**Cantao Wu[1], Yi Cai[1,2], Liuwu Li[1], Jiexin Wang[1,2],***

[1]School of Software Engineering, South China University of Technology
[2]Key Laboratory of Big Data and Intelligent Robot (South China
University of Technology) Ministry of Education
taotaotao0412@gmail.com
{ycai, jiexinwang}@scut.edu.cn
liuwu.li@outlook.com

## Abstract

Referring Expression Comprehension (ReC) is a task that involves localizing objects in images based on natural language expressions. Most ReC methods typically approach the task as a supervised learning problem. However, the need for costly annotations, such as clear image-text pairs or region-text pairs, hinders the scalability of existing approaches. In this work, we propose a novel scene graph-based framework that automatically generates high-quality pseudo region-query pairs. Our method harnesses scene graphs to capture the relationships between objects in images and generate expressions enriched with relation information. To ensure accurate mapping between visual regions and text, we introduce an external module that employs a calibration algorithm to filter out ambiguous queries. Additionally, we employ a rewriter module to enhance the diversity of our generated pseudo queries through rewriting. Extensive experiments demonstrate that our method outperforms previous pseudo-labeling methods by about 10%, 12%, and 11% on RefCOCO, RefCOCO+, and RefCOCOg, respectively. Furthermore, it surpasses the state-of-the-art unsupervised approach by more than 15% on the RefCOCO dataset.

## 1 Introduction

Referring expression comprehension (ReC) is an important visual-linguistic task that aims to locate objects in images based on given textual referring expressions. Referring expressions, such as "give me that apple in the basket," are commonly used in social communication. The ability to accurately comprehend such expressions in real-world scenarios is also essential for robots and other intelligent agents to enable natural interactions (Fang et al., 2015; Qi et al., 2020). In recent years, considerable progress has been made in ReC task, particularly for fully-supervised models (Chen et al., 2018b;

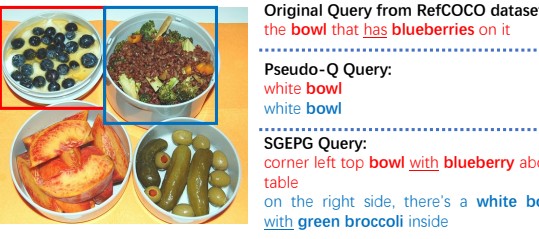

Figure 1: Example of queries generated by Pseudo-Q and SGEPG methods, based on two different objects in an image from the RefCOCO dataset. Note that the query generated by SGEPG without rewriting is shown in red, while the query generated by SGEPG with rewriting is shown in blue.

Deng et al., 2021; Yang et al., 2022; Ye et al., 2022) trained on manually annotated region-query data. However, constructing large-scale ReC datasets, especially for textual queries referring to the objects, is rather time-consuming. To obtain the clear textual queries, annotators need to provide specific visual information about the referred object, such as its category and attributes. Moreover, due to the presence of multiple similar and confusing objects in images, queries often require additional details to distinguish the referred object, such as spatial information (left, right), object state (running, jumping), and object relationships (e.g., person holding ball). Although some weakly-supervised methods (Chen et al., 2018a; Datta et al., 2019; Gupta et al., 2020) partially alleviate the annotation burden, they still rely on paired image-query data, limiting their applicability in the real-world scenes.

To address the annotation challenges, several works (Yeh et al., 2018; Wang and Specia, 2019; Jiang et al., 2022) have explored unsupervised approaches that do not rely on paired annotations. Nonetheless, these approaches either employ statistical hypothesis testing, make simple assumptions, or only investigate the shallow relation between objects, leading to poor performance in complex scenes. For instance, Pseudo-Q (Jiang et al., 2022),

---

*Corresponding author

which adopts a method of generating pseudo-labels to train a supervised model. It utilizes an offline object detector to extract salient objects from images and generates pseudo queries using predefined templates with object labels and attributes. However, Pseudo-Q has two limitations: (1) It fails to capture deep relation information among objects in images, leading to pseudo queries that only partially cover the semantic space of referring expressions and deviate significantly from human expressions. (2) The absence of a validation mechanism for the generated pseudo queries introduces the possibility of producing incorrect instances. Both limitations can lead to a degradation in model performance.

In this paper, we present SGEPG (Scene Graph Enhanced Pseudo-Query Generation), an unsupervised framework designed to generate high-quality region-query pairs. SGEPG leverages the generated scene graph to extract rich semantic information about objects and capture their complex relationships, enabling the selective generation of region-query pairs from the input image. We next introduce a straightforward yet effective strategy based on the scene graph, which helps filter out ambiguous queries. It is crucial to address ambiguity as unclear region-text pairs can yield incorrect supervisory signals, which ultimately degrade the performance of the ReC model. Additionally, we propose a module to rewrite some of the generated queries. This rewriting process enhances the diversity of the generated pseudo queries while also making them more akin to human expressions. As depicted in Figure 1, compared to Pseudo-Q, the SGEPG method generates pseudo queries that are both unambiguous and exhibit higher quality. Moreover, the generated queries by SGEPG also demonstrate more complex inference paths, such as the paths from "bowl" to the "blueberries".

Specifically, the SGEPG framework comprises four essential components: *Core Scene Graph Generation*, which is responsible for obtaining an informative core graph to capture object relations; *Non-Ambiguous Query Generation*, which focuses on generating unambiguous queries; *Diversity Driven Query Rewriter*, which is designed to rewrite some of the generated queries, aiming to increase the diversity and alleviate the differences between pseudo-queries and human expressions; *Visual-Language Module*, which aims to capture the contextual relationship between the image regions and corresponding pseudo queries.

The experimental results demonstrate the effectiveness of our method, which significantly reduces the manual labeling cost by about 40% on the RefCOCO dataset, with only a slight decrease in performance compared to the fully supervised setting. In addition, our proposed method consistently outperforms the state-of-the-art unsupervised approach by about 10-13 points on the RefCOCO (Yu et al., 2016), RefCOCO+ (Yu et al., 2016), and RefCOCOg (Mao et al., 2016) datasets. Our main contributions can be summarized as follows:

- We propose the first scene graph-based pseudo-labeling method for unsupervised ReC task, which effectively represents the key visual features of different objects and captures their relationships, resulting in region-query pairs that express richer relational information.

- To address the issue of generating ambiguous queries that yield incorrect supervisory signals and degrade performance, we introduce a disambiguation module based on the scene graph structure to further align the region-query modalities. Additionally, we propose rewriting some queries to increase the diversity and make them more closely resemble human expressions.

- Experimental results demonstrate that SGEPG not only effectively alleviates the reliance on manual annotation, but also outperforms counterparts and achieves comparable performance to large pre-trained vision-language methods.

## 2 Related Work

### 2.1 Referring Expression Comprehension

The task of Referring Expression Comprehension (ReC) plays a crucial role in applications such as robot navigation and visual question answering. ReC methods can be roughly classified into three types: fully supervised (Deng et al., 2021; Yang et al., 2022), weakly supervised (Gupta et al., 2020; Liu et al., 2019a, 2021; Sun et al., 2021), and unsupervised (Jiang et al., 2022; Subramanian et al., 2022; Wang and Specia, 2019; Yeh et al., 2018). Although fully supervised methods have achieved state-of-the-art results in the ReC task (Kamath et al., 2021; Yang et al., 2022), they heavily rely on vast and expensive manually annotated region-query pairs for training. In contrast, weakly supervised methods aim to alleviate this reliance by training the model using only image-text pairs. Most weakly supervised methods (Chen et al., 2018a; Liu et al., 2021) use object detectors to obtain po-

tential regions corresponding to the expressions, treating them as alternatives to missing bounding boxes for training the model. However, annotating the expressions in ReC datasets is the most time-consuming and labor-intensive process.

To address the need for manual annotations and reduce annotation costs, there is a growing interest in unsupervised methods (Jiang et al., 2022; Subramanian et al., 2022; Wang and Specia, 2019; Yeh et al., 2018). For instance, Wang and Specia (2019) employ multiple detectors (e.g., Faster-RCNN (Girshick, 2015)) to detect objects, scenes, and colors in images. By comparing the similarity between visual regions and the nouns in the query, the corresponding region related to the query can be identified. ReCLIP, proposed by Subramanian et al. (2022), introduces the use of CLIP (Radford et al., 2021) to assign scores to various image regions based on their similarity to the query. The region with the highest similarity score is then selected as the final result. While these methods leverage prior knowledge to analyze the alignment between visual regions and text, the inference speed is relatively slow due to the encoding and scoring of image regions and text in each step. One particular unsupervised approach, Pseudo-Q (Jiang et al., 2022), stands out by considering using pseudo-labelling techniques. The generated pseudo-labels can be employed to train supervised ReC models, allowing the models to acquire grounding abilities without the requirement for manually labeled data. However, Pseudo-Q fails to capture deep relation information among objects in images and does not address the ambiguity issue, leading to poor quality queries, as shown in Figure 1.

Unlike the existing pseudo-labeling methods (Cheng et al., 2021; Feng et al., 2019; Jiang et al., 2022), SGEPG incorporates scene graphs, enabling the generation of queries with inferred paths and rich relational information.

## 2.2 Scene Graph

Scene graphs contains structured semantic information about images, including object categories, attributes, and pairwise relationships. They can provide valuable prior information and have proven useful for various tasks like image captioning (Yang et al., 2019; Li and Jiang, 2019; Zhong et al., 2020). For instance, Li and Jiang (2019) argue that previous image caption approaches, which treat each object individually, lack structured repre-

sentation that provides important contextual clues. Therefore, they propose utilizing triples (subject, relation, object) from scene graphs as additional semantic information, along with visual features, as input to generate caption. Yang et al. (2019) introduce a Scene Graph Auto-Encoder module and incorporate inductive language bias into an encoder-decoder image captioning framework to produce more human-like captions.

The task of scene graph generation (Dai et al., 2017; Xu et al., 2017; Zhang et al., 2022; Zellers et al., 2018; Tang et al., 2019) aims to obtain a complete graph structure representation of an image, where nodes represent object categories and edges indicate relationships between objects (e.g., person-riding-bike). Early works on scene graph generation focused on local visual information and overlooked contextual information in images. To address this issue, Xu et al. (2017) proposed an end-to-end model that improves visual relationship prediction by iteratively propagating contextual information across the scene graph's topology. Furthermore, most scene graph generation models struggle with data distribution issues, such as long-tail distributions. This poses challenges in accurately predicting frequent relationships that lack informative details (e.g., on, at), limiting the applicability of scene graph models in practical tasks. To address this, Zhang et al. (2022) propose Internal and External Data Transfer. This approach automatically generates an enhanced dataset, mitigating data distribution problems and providing more coherent annotations for all predictions.

## 3 Methodology

In this section, we present the SGEPG framework, which aims to automatically generate region-query pairs from given images and train models to tackle ReC task using these pseudo labels. As depicted in Figure 2, the SGEPG framework consists of four key components: *Core Scene Graph Generation*, which is responsible for obtaining an informative core graph to capture object relations; *Non-Ambiguous Query Generation*, which focuses on generating unambiguous queries; *Diversity Driven Query Rewriter*, which is designed to rewrite some of the generated queries, aiming to increase the diversity and alleviate the differences between pseudo-queries and human expressions; *Visual-Language Module*, which aims to capture the contextual relationship between the image re-

gions and corresponding pseudo queries. Note that in this work, we mainly focus on the first three modules, with the objective of generating pseudo-labels of good quality for training visual-language models designed for the ReC task. To accommodate space limitations, certain methodological details may have been omitted. For a more comprehensive understanding, please refer to Appendix A for detailed information regarding the methods employed in this study.

## 3.1 Core Scene Graph Generation

The objective of this module is to obtain an informative core graph that accurately describes the objects in the image and their relationships. The scene graph, denoted as $G = (V, E)$, is a directed graph consisting of vertices $V$ representing object instances characterized by bounding boxes with object category labels, and edges $E$ representing relationships between objects. Each edge $e_i = (v_s, v_o, r_i)$ indicates a relation type $r_i$ between subject $v_s$ and object $v_o$.

Firstly, we utilize scene graph generation techniques to construct a preliminary scene graph for a given image. Specifically, we employ the model proposed by Zhang et al. (2022), as discussed in Section 2.2. However, the generated preliminary scene graph may not be complete due to the limitations of current scene graph approaches. These approaches often provide coarse and insufficient object categories in the generated graph, represented by generic labels such as "flower" instead of specific labels like "rose".[1] Furthermore, the generated scene graph only contains the relationship information between objects and fails to capture attribute details of each object, such as the color of a rose. Thus, the preliminary scene graph lacks specific information about object attributes.

To address the aforementioned limitations, we expand the preliminary graph by introducing additional nodes and edges to construct a complete scene graph. Following the insights from (Jiang et al., 2022), the referring expressions typically involve three crucial aspects: *category*, *attributes* and *spatial information*. Additionally, humans naturally combine these visual semantic cues when referring to specific target objects, particularly in complex scenes with multiple objects of similar classes, reducing ambiguity in the referring expres-

sions. Therefore, we next extract these three key aspects from the image:

1. Category. Inspired by two stage ReC methods (Yu et al., 2018), we employ an object detector to obtain object proposals (e.g., "skier" in Figure 2).

2. Attribute. To extract attributes like *material* (e.g., wooden), *color* (e.g., red), and *status* (e.g., standing), we utilize an attribute classifier (Anderson et al., 2018). However, existing models have a constraint in extracting only the most confident attribute for each object, even though an object can have multiple attributes.

3. Spatial Information. Humans often use spatial information to refer to an object, such as "right apple", "top guy". We obtain spatial information by comparing the center coordinates of object bounding boxes. Specifically, we derive horizontal spatial information (e.g., left, right) and vertical spatial information (e.g., top, bottom) by comparing the object center coordinates along these dimensions.

After extracting the above three information, we first update the categories of the vertices in the preliminary scene graph with the obtained category information. Next, we introduce attribute nodes and spatial nodes into the scene graph, representing the attribute information and spatial relationships with the corresponding object nodes, that are denoted as *[Attr]*, *[Spatial]*, respectively. As a result, we generate an informative core scene graph that encompasses the three crucial pieces of information, as illustrated in the lower left corner of Figure 2.

## 3.2 Non-Ambiguous Query Generation

The core scene graph captures the rich semantic information of the entire image, which can be utilized to generate queries by populating its information into designed templates.[2] However, using the entire scene graph may result in queries that are too complex and contain excessive specific information for referring the objects. In ReC tasks, queries are typically concise and only describe partial information relevant to the objects being referred to, rather than providing an overall description of the scene. To address this, we focus on subgraphs that contain the nodes corresponding to the objects of interest. These subgraphs capture partial information about the objects, which can be leveraged to obtain concise pseudo queries.

However, the ambiguity issue may arise if there are similar subgraphs present in the scene graph.

---

[1]Note that the model proposed by Zhang et al. (2022) only supports 150 classes.

[2]Please refer to Appendix B for the designed templates.

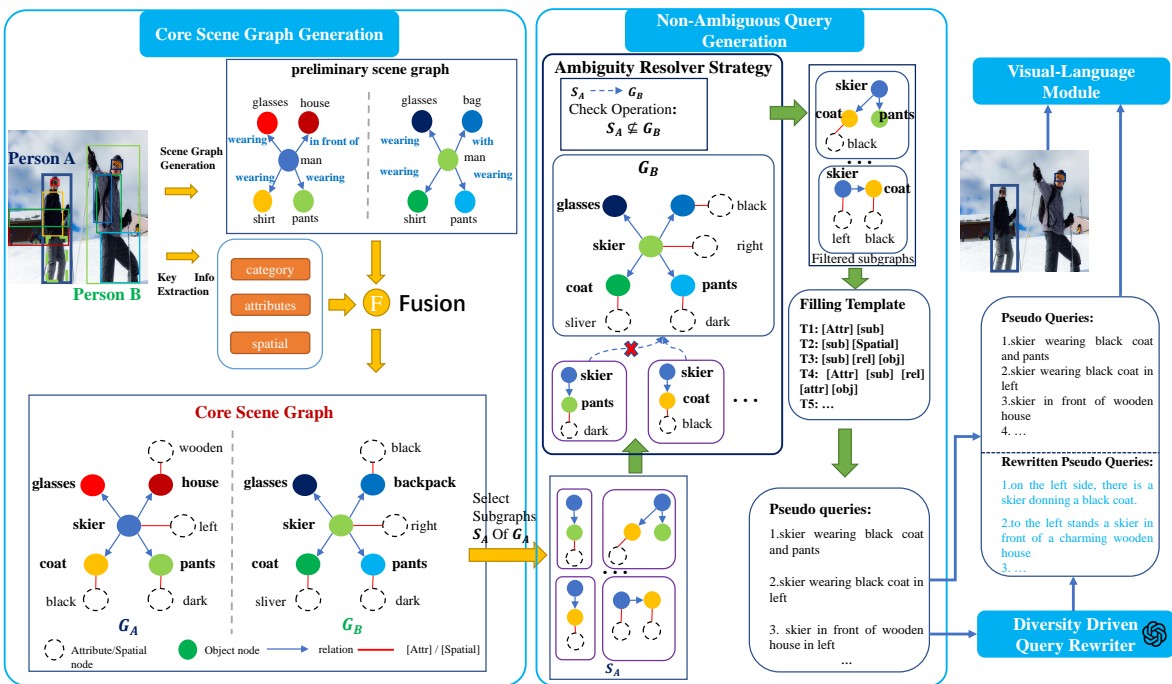

Figure 2: The framework of SGEPG, which consists of four modules: Core Scene Graph Generation, Non-Ambiguous Query Generation, Diversity Driven Query Rewriter, and Visual-Language. The diagram also illustrates how SGEPG generates queries related to Person A.

An ambiguous query can refer to multiple objects in the image, which might lead to incorrect supervision signals detrimental for effective learning of visual-language module. Existing pseudo query-based models, such as Pseudo-Q, struggle with this issue and may generate ambiguous queries that refer to different objects, as depicted in Figure 1. To address this issue, we propose a simple yet effective `Ambiguity Resolver Strategy` based on the characteristics of the scene graph. The main idea is that a query referring to an object is non-ambiguous if its corresponding subgraph is not a subgraph of any other subgraphs representing different objects in the scene graph. Specifically, let $G_A$ and $G_B$ denote the graphs of objects $A$ and $B$, respectively, and $S_A$ represent the selected subgraph from $G_A$ for generating the query about $A$. To resolve ambiguity, $S_A$ should satisfy the condition $S_A \nsubseteq G_B$. For instance, in the left corner of the Non-Ambiguous Query Generation module on the right side of Figure 2, the subgraph representing "skier-pants-dark" of "Person A" is removed as it is identical to a subgraph of "Person B".

### 3.3 Diversity Driven Query Rewriter

To address the limited diversity and fluency issues of pseudo queries generated by directly filling sub-graphs into predefined templates, we introduce a

rewriter component. It aims to to enhance the diversity and bridge the gap with human expressions, while preserving semantic consistency. To achieve this, GPT-3.5, a poweful large pre-trained language model is utilized to rewrite the queries. GPT-3.5 also has the capability to incorporate additional prior knowledge into queries. For example, it may rewrite "giraffe" as "long-necked animal" associating the abstract concept of a "long-necked" with the entity of "giraffe". This integration of prior knowledge helps the SGEPG generate more accurate and meaningful queries. Furthermore, we introduce an important hyperparameter $\alpha$, which determines the proportion of generated pseudo queries that will undergo the rewriting process.

### 3.4 Visual-Language Module

In this module, our objective is to train a visual-language model that learns the mapping between image regions and textual descriptions, using pseudo-labels obtained from the previous modules. To achieve this, we adopt a similar architecture to VLTVG (Yang et al., 2022), a visual-language model designed for the ReC task through supervised learning. The VLTVG model consists of four components: a visual encoder, a language encoder, a fusion module, and a multi-stage cross-modal decoder. To extract visual features from the given

image, we employ DETR (Carion et al., 2020) as the visual encoder within our model. For the textual modality, we utilize BERT (Devlin et al., 2018) as the language encoder to obtain feature representations of the corresponding expressions. These visual and textual representations are then fused using a fusion module. Finally, the fused features are passed through a multi-stage cross-modal decoder, which generates the coordinates for the final predicted bounding box for the referred object in the image. Note that while we have specifically utilized this visual language model architecture, it can be replaced by any other supervised ReC models.

## 4 Experiment

### 4.1 Datasets and setups

We evaluate SGEPG on three datasets: **RefCOCO**, **RefCOCO+** and **RefCOCOg**. The images in these datasets are from the MSCOCO (Vinyals et al., 2016) and we follow the same train/val/test splits from (Subramanian et al., 2022). The number of training images for these three datasets is 16994, 16992, and 24698, respectively. Compared to RefCOCO, the queries in RefCOCO+ do not include spatial words (e.g. "left", "top") to describe objects, and RefCOCOg queries are typically longer and involve more interactions between objects. The referred objects in testA split are all human, the testB split comprises only non-human objects, while val split does not have such restriction.

For each image in the RefCOCO, RefCOCO+, and RefCOCOg datasets, we select the top 3 objects based on their confidence scores and generate 3 queries for each selected object. The hyperparameter $\alpha$ for the Rewriter is set to 50%. The visual-language component of our proposed model is trained independently for 30 epochs using the generated pseudo-labels of the images from each dataset, with the remaining hyperparameter settings consistent with those reported in the VLTVG paper. The evaluation of the trained model is conducted on the respective test set of each dataset. Thus, during the training stage of our approach, we do not utilize any manually annotated datasets, i.e., the human annotated image region-query pairs, which are solely used for evaluation purposes.

### 4.2 Baselines

We compare the performance of our SGEPG approach with several existing methods, including unsupervised methods such as ReCLIP, GradCAM

(Subramanian et al., 2022), CPT (Yao et al., 2021), and Pseudo-Q(Jiang et al., 2022) as well as weakly-supervised methods like VC (Zhang et al., 2018), ARN (Liu et al., 2019a), KPRN (Liu et al., 2019b), and DTWREG (Sun et al., 2021). We also include the random guess results and the state-of-the-art results achieved by MDETR (Kamath et al., 2021). It is worth noting that Pseudo-Q, like our SGEPG approach, also relies on generating pseudo queries, as mentioned earlier. However, Pseudo-Q employs a different visual-language model than VLTVG (Yang et al., 2022), which we have chosen as our visual language model. To ensure a fair comparison, we train a VLTVG model using the region-text pairs generated by Pseudo-Q, and refer to it as *Pseudo-Q-adaptive* in our evaluation.

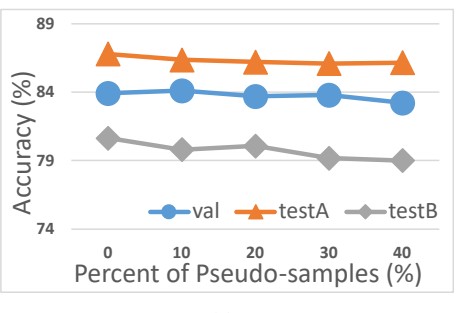

(a)

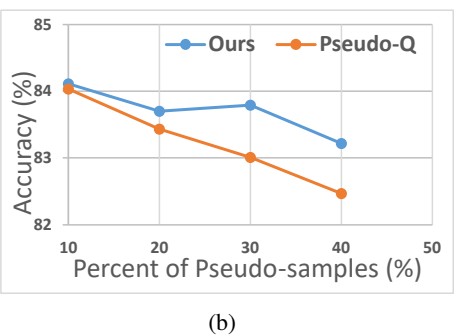

(b)

Figure 3: Experiments of reducing the manual labeling cost. Figure (a): results of our method on three splits of RefCOCO; Figure (b): comparision results between our method and Pseudo-Q on val split of RefCOCO.

### 4.3 Main Results

The results on RefCOCO, RefCOCO+, and RefCOCOg are presented in Table 1. We begin by comparing our proposed SGEPG model with the unsupervised methods: ReCLIP, and Pseudo-Q. We observe that our proposed SGEPG model demonstrates superior performance. In comparison to ReCLIP, the state-of-the-art unsupervised method which relies on pre-trained visual-language models, SGEPG surpasses ReCLIP by a remarkable 15% on

| method | Sup | RefCOCO | | | RefCOCO+ | | | RefCOCOg | |
|---|---|---|---|---|---|---|---|---|---|
| | | val | testA | testB | val | testA | testB | val | test |
| Random | - | 15.73 | 13.51 | 19.20 | 16.29 | 13.57 | 19.60 | 18.12 | 19.10 |
| Supervised SOTA | Full | 87.51 | 90.40 | 82.67 | 81.13 | 85.52 | 72.96 | 83.35 | 81.64 |
| VC(Zhang et al., 2018) | Weak | - | 33.29 | 30.13 | - | 34.60 | 31.58 | - | - |
| ARN(Liu et al., 2019a) | | 34.26 | 36.43 | 33.07 | 34.53 | 36.01 | 33.75 | - | - |
| KPRN(Liu et al., 2019b) | | 35.04 | 34.74 | 36.98 | 35.96 | 35.24 | 36.96 | - | - |
| DTWREG(Sun et al., 2021) | | 39.21 | 41.14 | 37.72 | 39.18 | 40.10 | 38.08 | - | - |
| **CPT**(Yao et al., 2021) | No | | | | | | | | |
| CPT-Blk w/ VinVL | | 26.9 | 27.5 | 27.4 | 25.4 | 25.0 | 27.0 | 32.1 | 32.3 |
| CPT-Seg w/ VinVL | | 32.2 | 36.1 | 30.3 | 31.9 | 35.2 | 28.8 | 36.7 | 36.5 |
| **CLIP** | No | | | | | | | | |
| GradCAM | | 44.65 | 53.49 | 36.19 | 49.41 | 59.66 | 38.62 | 52.29 | 51.28 |
| ReCLIP | | 54.04 | 58.60 | 49.54 | 55.07 | 60.47 | **47.41** | 60.85 | **61.05** |
| **Pseudo-labeling** | No | | | | | | | | |
| Pseudo-Q(Jiang et al., 2022) | | 56.02 | 58.25 | 54.13 | 38.88 | 45.06 | 32.13 | 46.25 | 47.44 |
| Pseudo-Q-adapted | | 59.44 | 61.87 | 56.47 | 42.76 | 48.27 | 36.90 | 50.35 | 49.68 |
| **SGEPG (Ours)** | | **69.61** | **72.34** | **65.67** | **55.21** | **60.67** | 46.76 | **61.33** | 60.61 |

Table 1: Comparison with ReC methods on RefCOCO/+/g in terms of top-1 accuracy (%). **Bold** indicates the best performance of the column except for the supervised method. Sup column indicates the method category: Full, Weak, No represent fully supervised, weakly supervised and unsupervised, respectively.

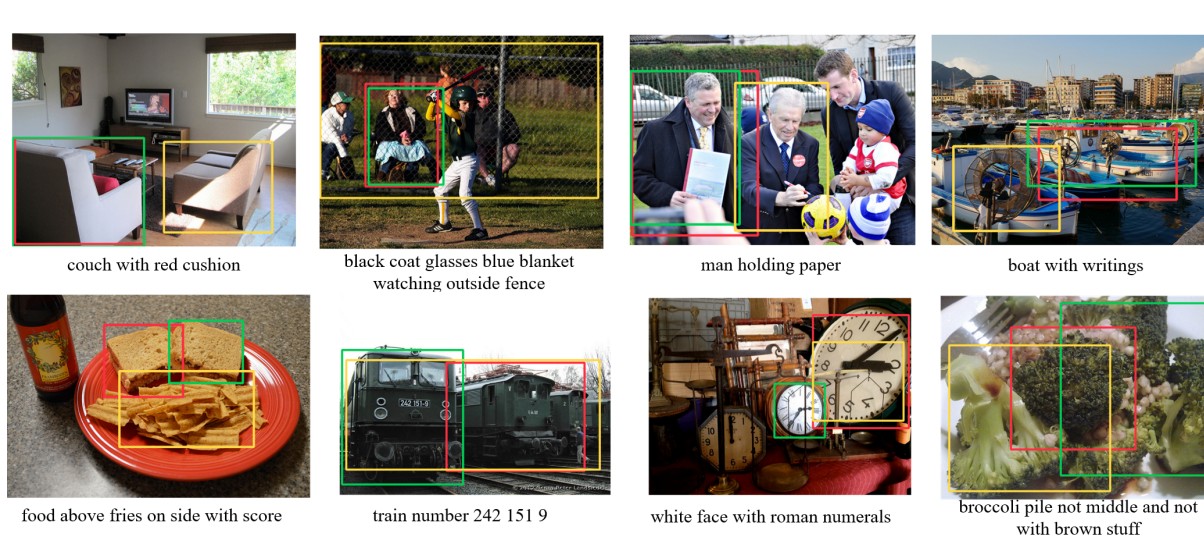

couch with red cushion

black coat glasses blue blanket watching outside fence

man holding paper

boat with writings

food above fries on side with score

train number 242 151 9

white face with roman numerals

broccoli pile not middle and not with brown stuff

Figure 4: Visualization examples of detection results. The first row of images displays the visualized detection results of both SGEPG and Pseudo-Q. The second row showcases four instances where SGEPG encountered detection failures, with the corresponding results from Pseudo-Q included for comparison. The green bounding boxes and queries are ground truth and the red bounding boxes and yellow bounding boxes are the detection results of SGEPG and Pseudo-Q, respectively.

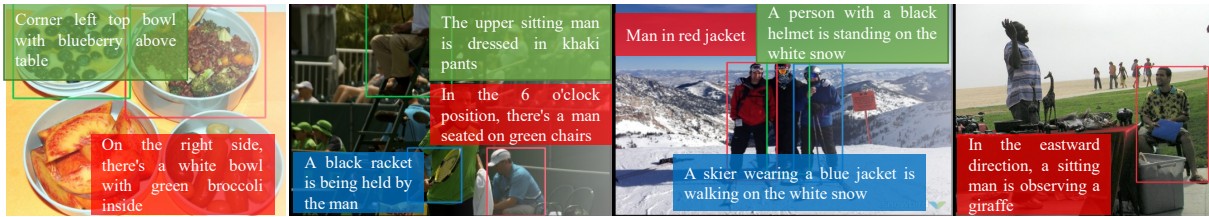

Figure 5: The visualization results of pseudo-labels generated by SGEPG. Different colored queries corresponding to different objects.

| | RefCOCO | | | RefCOCO+ | | |
|---|---|---|---|---|---|---|
| | val | testA | testB | val | testA | testB |
| w/o SG | 63.45 | 63.9 | 63.47 | 43.8 | 45.81 | 44.14 |
| w/o ARS | 65.24 | 63.81 | 64.65 | 48.5 | 50.47 | 46.02 |
| w/o DDRQ | 68.82 | **72.46** | 64.99 | 53.78 | 58.26 | 46.14 |
| **SGEPG** | **69.61** | 72.34 | **65.67** | **55.21** | **60.67** | **46.76** |

Table 2: Main ablation test results on the RefCOCO/+.

| replace rate ($\alpha$) | 0% | 25% | 37.5% | 50% | 75% |
|---|---|---|---|---|---|
| val | 53.78 | 54.29 | 54.94 | 55.21 | 55.33 |
| testA | 58.26 | 60.15 | 58.99 | 60.67 | 60.9 |
| testB | 46.14 | 46.41 | 46.7 | 46.76 | 46.8 |

Table 3: Rewriting ratio test results of DDQR on Ref-COCO+.

the RefCOCO dataset, while achieving comparable performance on RefCOCO+ and RefCOCOg. Regarding Pseudo-Q, which is also based on pseudo-labeling, our SGEPG model outperforms it by an average margin of 10% to 12.5%. This demonstrates that the queries generated by SGEPG for referring to objects can better ensure accurate mapping between the two modalities.

## 4.4 Efficiency Improvement of Manual Annotation

We then conduct experiments to assess the capability of SGEPG in reducing manual labeling costs. Following Pseudo-Q, we replace the manually annotated region-text pairs whose queries contain spatial relationships in RefCOCO with the pseudo-labels generated by SGEPG. The results, depicted in Figure 3(a), demonstrate the significant reduction in manual labeling costs achieved by SGEPG. When about 40% of the manually annotated region-text pairs are replaced with our generated pseudo-labels, there is only minimal impact on the model's performance. In addition, we compared SGEPG with Pseudo-Q in Figure 3(b), which indicates that our SGEPG model has the ability to generate superior quality pseudo-labels compared to Pseudo-Q.

## 4.5 Ablation Study

In this section, we perform extensive ablation studies on the RefCOCO and RefCOCO+ datasets to evaluate the effectiveness of the key aspects of SGEPG, including *scene graph (SG)*, *ambiguity resolver strategy (ARS)*, and *Diversity Driven Query Rewriter (DDQR)*. To evaluate the impact of the SG component, we compare the performance of the model without SG. In this scenario, the category, attribute, and spatial information of the detected

objects are directly used to fill the templates and generate pseudo queries. Moreover, as the ARS relies on the scene graph, the model without SG cannot utilize this strategy to address ambiguity. For the models without ARS or DDQR, we either exclude the strategy or remove the DDQR component directly from the SGEPG model. The results of this ablation study are presented in Table 2.

Firstly, when comparing the performance of the model without SG to the complete SGEPG, we observe that the complete SGEPG achieves absolute performance improvements of 6.16% and 11.41% on the validation sets of RefCOCO and RefCOCO+, respectively. This also indicates that the SG can effectively improve the model's ability to align visual and textual information, especially on challenging datasets like RefCOCO+. Secondly, the complete SGEPG also outperforms the model without the ARS, with absolute improvements of 8.53% and 10.2% on the RefCOCO/+ testA, respectively. Finally, we observe that when the DDQR module is removed, the model's performance decrease across both datasets. The above findings demonstrate that the modules proposed by us can provide more accurate supervision signals to the ReC model and improve its performance.

## 4.6 Effect of Different Replacing Ratios in Rewriter

In this section, we conduct experiments to examine the impact of rewriting ratios $\alpha$ in the DDRQ. We perform rewriting on pseudo queries with $\alpha$ values of 25%, 37.5%, 50%, and 75%. The final results are presented in Table 3. The results indicate that as the rewriting ratio increases, the model's performance improves. This finding suggests that incorporating more diverse expressions can alleviate the discrepancy between pseudo queries generated by artificial templates and human expression. However, there is a noticeable diminishing return effect: the more sentences are rewritten, the training cost increases linearly while the performance gain becomes limited. Therefore, we strike a balance between training cost and model performance, and find that a rewriting ratio of 50% is appropriate.

## 4.7 Qualitative Analysis

**Qualitative examples.** To validate that our approach can provide queries with rich relational information and enhance the model's perception of relationships, we conducted visualizations of some generated examples in Figure 5. Further-

more, we performed visual comparisons with the results from Pseudo-Q on selected cases, directly comparing them (first row of Figure 4). We observe that SGEPG effectively captures the complex relational information among objects in the scene, while Pseudo-Q tends to randomly select an object from the query's nouns as the detection object. For instance, in the third example in first row, Pseudo-Q generates detection results solely based on the word "Person" in the sentence, completely disregarding the "holding" relationship and the "paper" object. This also suggests that SGEPG is capable of better capturing the relationships between objects.

**Failure case.** Additionally, while second row of Figure 4 presents cases where SGEPG encountered prediction errors that can primarily be attributed to the two limitations of the scene graph. Firstly, the scene graph may lack the ability to recognize text present in images. Secondly, the scene graph generation model may struggle in complex scenes, resulting in inaccurate scene graph generation.

## 5 Conclusion

In this paper, we propose SGEPG, a novel pseudo-labeling ReC method that leverages scene graphs to improve the generation of pseudo queries. By utilizing the scene graph, SGEPG captures object relationships and generates pseudo queries that are rich in relation information, thereby improving the training of ReC models. Additionally, we propose a simple yet effective strategy to filter out ambiguous generated queries. Our method outperforms previous pseudo-labeling approaches on three benchmark datasets and achieves comparable or superior performance compared to other unsupervised ReC methods. Furthermore, SGEPG offers the advantage of significantly reducing manual labeling costs while maintaining the model's performance.

## Limitations

While our method has demonstrated excellent performance on three datasets, there are still some limitations. Firstly, the quality of the generated results relies on the accuracy of the scene graph generation. In complex scenes, there is no guarantee that all objects and their relationships can be correctly identified, which may impact the effectiveness of our approach. Secondly, the capabilities of the scene graph are limited in terms of object detection. It can only detect a specific set of objects and may fail to identify certain objects such as the sky or

sea. Consequently, the generated pseudo queries may not adequately cover these object classes, potentially affecting the comprehensiveness of the generated queries.

## 6 Acknowledgement

This work was supported by the National Natural Science Foundation of China (62076100), Fundamental Research Funds for the Central Universities, SCUT (x2rjD2230080), the Science and Technology Planning Project of Guangdong Province (2020B0101100002), CAAI-Huawei MindSpore Open Fund, CCF-Zhipu AI Large Model Fund.

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

# A  More Details of the Scene Graph Generation

## A.1  Spatial

In this section, we will elaborate on some additional details regarding the extraction of spatial information. In addition to the four cardinal directions (up, down, left, and right) that we are familiar with, we have incorporated some extra spatial descriptions to help the model better learn the concept of "spatial".

Firstly, we introduce the concept of "corner". Sometimes, humans use the term "corner" to refer to objects that are located in the corners of an image. Therefore, when extracting the spatial information of objects, we also determine whether the object is positioned at the edges or corners of the image. In this work, we consider an object to have the "corner" attribute if it satisfies the following conditions: Firstly, the object's bounding box should not be located in the center of the image. Secondly, the object's bounding box should have at least two edges that are positioned at the image's edges. Objects that meet these criteria are assigned the additional "corner" attribute.

Secondly, we have observed that people often use the term "alone" to refer to an object that appears to be "solitary". Therefore, when there are multiple instances of a certain object category in an image (typically three or more), we employ clustering algorithms(Ball and Hall, 1965) based on coordinates and object bounding box sizes to attempt to divide them into two groups. If one of the groups contains exactly one object, we consider that object to be "alone" and assign it the additional "alone" attribute.

## A.2  Scene Graph Cleaning

While we can utilize scene graph generation models to extract scene graphs from images, the obtained scene graphs may contain errors due to limitations in model performance. Although most incorrectly predicted relations can be filtered out based on confidence scores, some contradictory results may still remain. For example, a scene graph may contain two conflicting triples such as "boy-sitting at-chair" and "boy-standing on-floor," which contradicts common sense as a person cannot be both "sitting" and "standing." In such cases, we only retain the triple with the highest confidence level.

## B  Designed Template

Due to the fact that scene graphs are graph structures while natural language is a sequential text, it is necessary to find an appropriate method to convert graph structures into text sequences. In this paper, we employ a method that involves populating a manually designed template with information extracted from the scene graph. The scene graph encompasses four main types of information: 1. Object categories, 2. Object attributes, 3. Relationships between objects and 4.spatial. Therefore, the elements to be filled in the manually designed template primarily consist of these three types of information. Here are the main templates used in this paper:

- [attr] [sub]

- [sub] [rel] [obj]

- [sub] in/on [spatial] {corner | alone}

- [attr] [sub] in/on [spatial] {corner | alone}

- [attr] [sub] [rel] [obj] in/on [spatial]

- [attr] [sub] [rel] [obj] and [rel] [obj]

- [attr] [sub] [rel] [attr] [obj] and [rel] [attr] [obj]

- [attr] [sub] [rel] [attr] [obj] and [rel] [attr] [obj] in [spatial] {corner | alone}

In this context, "attr" refers to the attribute of an object, "sub" represents the subject, "obj" represents the object, "spatial" indicates the spatial, and "rel" represents the relationship between subject and object. Additionally, "corner" and "alone" are optional and indicate whether the object is located in the corner of the image and whether it is "alone," respectively. By filling in the scene graph information into the templates, we can obtain pseudo queries generated based on the scene graph. Please note that the word order may vary, and "spatial" could be placed at the beginning of the sentence.

## C  Prompt of Rewriter

We use GPT3.5 (ChatGPT) to rewrite the pseudo queries we have constructed. We employ a prompt to communicate to GPT3.5 that our task objective is to rewrite sentences, with each sentence being rewritten three times. The following is the specific prompt we use, along with an example:

**Prompt**:*"Your goal is to act as human-being to generate diversified, accurate and semantically rich referring expressions according to a description of the referred object. The requirement of generated expression: The category, color, position, texture, relationship with other objects or others segments (entities) may be reorganized with some alternative expressions. I will give you some descriptions,and then you generate 3 referring expressions for per description. You should return a dict in which the key is the description and the values are the 3 expression involed in a list, the format looks like {"Description": "xxxxx", "Values": {"1": "xxxxx", "2": "xxxxx", "3": "xxxxx"} }. Don't leave out any sentences. Now, begining your generation following the goal, requirement and format:"*

**Input**: *center tallest man in glasses*

**Output**: *{ "Description":"Center tallest man in glasses": {"1":"At the center, the tallest man wearing glasses is present.", "2":"Glasses-wearing man who is the tallest is in the center.", "3":"The center position is occupied by a man, who is the tallest and wears glasses."}}*

**Input**: *corner left top bowl with blueberry above table*

**Output**: *{ "Description": "corner left top bowl with blueberry above table", "Values": { "1": "The blueberry-filled bowl is positioned in the top left corner, above the table.", "2": "Above the table, there's a bowl in the top left corner, and it contains blueberries.", "3": "In the top left corner, there's a bowl placed above the table, and it has blueberries inside." } }*

It is important to note that not every rewriting attempt is successful. There are instances where GPT-3.5 may fail to rewrite or generate unexpected statements, such as outputs that do not meet the expected format or provide insufficient information. For such cases where rewriting is not possible, we simply skip them and proceed without making any changes.

## D  Additional Experiments

### D.1  Effect of Expressions Number per Object and Object Number per Image

In this section, we investigate the impact of the number of generated pseudo queries for each object, as well as the number of pseudo-labeled objects generated per image, as shown in Table 4. Please note that in this experiment, by default, 4 objects are selected per image to generate pseudo-

| | Expression number per object | | | | Object number per image | | | | |
|---|---|---|---|---|---|---|---|---|---|---|
| | 1 | 2 | 3 | 4 | 1 | 2 | 3 | 4 | 5 |
| SGEPG | 66.17 | 68.21 | 68.6 | 68.81 | 62.3 | 67.61 | 68.82 | 68.6 | 68.02 |

Table 4: Results of different expressions numbers per object and different object numbers per image on RefCOCO.

| | RefCOCO | | | RefCOCO+ | | |
|---|---|---|---|---|---|---|
| | val | testA | testB | val | testA | testB |
| preliminary scene graph | 36.23 | 45.31 | 25.32 | 37.21 | 46.11 | 26.55 |
| Core scene graph (SGEPG) | 69.61 | 72.34 | 65.67 | 55.21 | 60.67 | 46.76 |

Table 5: preliminary scene graph generation results on RefCOCO and RefCOCO+.

labels, and 3 statements are generated for each object.

Intuitively, generating a larger number of pseudo queries for each object allows for a broader coverage of the semantic space. The experimental results demonstrate that increasing the number of expressions generated per object leads to improved performance, while insufficiently generated expressions result in limited coverage of the semantic space, leading to a decline in model performance. However, excessively generating expressions for each object can significantly increase training time and increase costs. Hence, to strike a balance between performance and training efficiency, we have set this value to 3 in our other experiments. Furthermore, as the number of objects increases, the number of generated pseudo queries also rises, leading to an improvement in our model's performance. However, it is important to note that our approach utilizes the confidence score from the object detector to select salient objects. Objects with low confidence are more prone to providing incorrect information regarding object categories and visual regions. Thus, when the number of objects is higher, the likelihood of introducing such erroneous information and negatively impacting the model's performance increases.

## D.2 Effect of Core Scene Graph

In order to assess the impact of scene graph generation quality and key information on the results of pseudo-label generation, we also attempted to generate pseudo-labels directly using a preliminary scene graph rather than a core scene graph. We performed the experiments on both the RefCOCO and RefCOCO+ datasets. In these experiments, we generated pseudo-labels directly using the preliminary scene graph without any further processing.

Experimental results are shown in Table 5. Notably, we observed that training the model solely with preliminary scene graphs, which are noisy and unprocessed, led to poor performance. This finding underscores the importance of accurate and refined core scene graphs in our approach. These core scene graphs, complemented with additional information from off-the-shelf object detectors and attribute classifiers, play a crucial role in enhancing the quality of generated pseudo-labels and ultimately improving the performance of the visual-language model.