# OpenReview forum: "Scene Graph Enhanced Pseudo-Labeling for Referring Expression Comprehension"
_EMNLP/2023/Conference — EMNLP 2023 Findings_

### Official Review · Reviewer_2nSK · 2023-08-02

**Typos Grammar Style And Presentation Improvements:** 1. Missing spaces between word and pu…
**Soundness:** 3

**Excitement:**

4: Strong: This paper deepens the understanding of some phenomenon or lowers the barriers to an existing research direction.

**Paper Topic And Main Contributions:**

This paper presents a pseudo-labeling approach to generate training examples for the Referential Expression Comprehension (ReC) task in an unsupervised manner using scene graphs. Specifically, the approach consists of three components to generate and decorate a core scene, filter out ambiguous queries, and rewrite the query for greater diversity. The main contributions of this paper are
1. the scene graph-based pseudo-labeling framework,
2. techniques such as the disambiguation module and the rewriting component to improve the quality of pseudo-labeling, and
3. quantitative results showing that the high quality of the data enables better model performance compared to previous unsupervised approaches.

**Questions For The Authors:**

Question A: Following the concern above, I am wondering how much performance gain on the RefCOCO dataset is gained by using the scene graph vs. extracting spatial information. Could you please provide metrics for the ablation study on the use of spatial information on the RefCOCO dataset?
Question B: The core scene graph generation flow is interesting with a preliminary graph generated with a model supporting a limited number of classes, and then decorated with more fine-grained categories, attributes, and spatial information with a separate extractive model. The paper cited the limitation of the scene graph generation model for the two-stage approach. Do you think that extending the capabilities of the scene graph generation model (to support additional classes and attributes/spatial information extraction) would provide better or similar results?

**Reasons To Accept:**

1. The scene graph-based pseudo-labeling approach is novel and is shown to outperform previous pseudo-labeling methods on standard datasets.
2. The choice of scene graph as well as the design of various techniques to further improve the performance of the approach is well motivated, described in detail, and the benefits are validated in the ablation studies.

**Reasons To Reject:**

How much performance improvement the approach can provide in more general cases is less clear. In particular, the RefCOCO dataset contains spatial words in the description that can be better captured by the extracted spatial information and not inherently validating the gain from using scene graphs or capturing relationships. As expected, we observe a larger performance gain using this approach on the RefCOCO dataset compared to the second place approach. For the other two datasets (RefCOCO+ and RefCOCOg), which are more complex and do not contain spatial relations, the performance of the system using the proposed approach is very close to the ReCLIP approach.

**Reproducibility:**

4: Could mostly reproduce the results, but there may be some variation because of sample variance or minor variations in their interpretation of the protocol or method.

**Reviewer Confidence:**

4: Quite sure. I tried to check the important points carefully. It's unlikely, though conceivable, that I missed something that should affect my ratings.

---

> ### Author Rebuttal · Authors · 2023-08-29
>
> Thank you for your constructive comments and we have carefully considered the important issues you raised.
>
> **(a)	How much performance improvement the approach can provide in more general cases is less clear. In particular, the RefCOCO dataset contains spatial words in the description that can be better captured by the extracted spatial information and not inherently validating the gain from using scene graphs or capturing relationships. As expected, we observe a larger performance gain using this approach on the RefCOCO dataset compared to the second place approach. For the other two datasets (RefCOCO+ and RefCOCOg), which are more complex and do not contain spatial relations, the performance of the system using the proposed approach is very close to the ReCLIP approach.**
>
> **A:** We appreciate your thorough analysis and concerns. We would like to address these points and provide further clarification:
>
> 1. We agree that handling spatial information play a crucial role in referring expression comprehension tasks. However, approaches like Pseudo-Q and ReCLIP have also introduced specific methods or modules to address spatial information. For example, Pseudo-Q incorporate spatial words into pseudo-labels, and ReCLIP employs a dedicated "Spatial Relation Resolver" module for handling spatial information. And we also found that RefCOCOg dataset also contains spatial information words and descriptions with redundant information.
> 2. In our experiments, we consistently outperform Pseudo-Q, which is also based on pseudo-labels, across various datasets. This demonstrates that our method, with the incorporation of scene graphs and the proposed improvements, is more effective in capturing object relationships, not limited to spatial information, and producing accurate pseudo-labels for training the visual-language model.
> 3. Our method's consistent performance across different dataset complexities is noteworthy. ReCLIP's unique pattern of performing well on complex datasets (RefCOCO+/g) and performing poor on simpler ones (RefCOCO) can be attributed to its origin in CLIP, which excels in handling complex textual scenes and struggles with shorter text and visually similar objects. On the other hand, our approach remains stable on complex datasets (RefCOCO+/g) and performs exceptionally well even on datasets with simpler textual descriptions (RefCOCO). This also suggests that our method's effectiveness extends beyond the specific linguistic characteristics of individual datasets.
> 4. In our ablation experiments (Section 4.2), we conducted comprehensive analyses of the influence of scene graphs on model performance. The results, presented in Table 2, showcase significant gains from scene graphs for both RefCOCO and RefCOCO+ datasets. This holds true even for RefCOCO+, which lacks spatial words. For instance, on the validation set of RefCOCO+, the inclusion of scene graphs improves the model's performance from 43.8 to 48.5, reflecting a substantial enhancement.
>
> **(b)	Following the concern above, I am wondering how much performance gain on the RefCOCO dataset is gained by using the scene graph vs. extracting spatial information. Could you please provide metrics for the ablation study on the use of spatial information on the RefCOCO dataset?**
>
> **A:** We appreciate your suggestion and have already conducted the experiments as suggested to assess the impact of using spatial information. We removed the utilization of spatial information while retaining the scene graph-based approach. The results clearly demonstrate a significant decrease in performance when spatial information is excluded, which underscores the importance of spatial information in the ReC domain.
>
> Moreover, it's worth noting that the consideration of spatial information is not unique to our approach but rather reflects the practical demands of the ReC task. Baselines like Pseudo-Q and ReCLIP also incorporate mechanisms to deal with spatial information, as mentioned in the first reponse. We hope these insights address your query and provide a clearer perspective on the importance of spatial information in our approach.
>
> |                       |       | RefCOCO |       |
> | --------------------- | ----- | ------- | ----- |
> |                       | val   | testA   | testB |
> | w/o spatial           | 45.43 | 53.76   | 34.60 |
> | with spatial（SGEPG） | 69.61 | 72.34   | 65.67 |
>
>
> **(c)	The core scene graph generation flow is interesting with a preliminary graph generated with a model supporting a limited number of classes, and then decorated with more fine-grained categories, attributes, and spatial information with a separate extractive model. The paper cited the limitation of the scene graph generation model for the two-stage approach. Do you think that extending the capabilities of the scene graph generation model (to support additional classes and attributes/spatial information extraction) would provide better or similar results?**
>
> **A:** Thank you for raising this interesting point. Expanding the capabilities of the scene graph generation model to encompass a broader range of object categories and attributes is indeed an appealing idea. This expansion could potentially lead to the capture of more accurate semantic information from images, resulting in higher-quality pseudo-labels and, in turn, improved model performance. However, there might be a potential issue here. Directly increasing the number of recognized object categories in the scene graph generation model could also introduce complexities during training. For instance, inferring relationships between a larger set of objects might become more intricate, potentially leading to convergence issues. Yet, it's worth noting that technology is continuously advancing, and our approach could directly benefit from the ongoing progress in the field of scene graph generation models.

---

### Official Review · Reviewer_G4b6 · 2023-08-07

**Soundness:** 4

**Excitement:**

4: Strong: This paper deepens the understanding of some phenomenon or lowers the barriers to an existing research direction.

**Paper Topic And Main Contributions:**

- This paper proposes scene graph enhanced pseudo-query generation (SGEPG) for referring expression comprehension. The proposed method shows significant performance improvements in RefCOCO, RefCOCO+, and RefCOCOg benchmark datasets.

**Questions For The Authors:**

- Question A: A preliminary scene graph is complemented with the additional information from the off-the-shelf object detector and attribute classifier and spatial information of the bounding box of each object. Have you conducted experiments using a preliminary scene graph, not utilizing a core scene graph? This ablation study would be helpful for readers to understand the impact of the quality of generated scene graphs in the pseudo-query generation process.
- Question B: This paper adopts GPT-3.5 to make pseudo queries to be diverse and fluent. An additional qualitative analysis of the rewriter would be needed to validate the effectiveness of the rewriting strategy.

**Reasons To Accept:**

- This paper is well-written and easy to follow. The motivation and design choice of the proposed method is simple and intuitive. The proposed method shows consistent performance improvement on the three benchmark datasets for referring expression comprehension.

**Reasons To Reject:**

- Although SGEPG is model-agnostic, this paper conduct experiments using a specific model, VLTVG, to validate the proposed method. The additional experiments using the other VLM would be helpful to emphasize the effectiveness of the scene graph-enhanced pseudo-labeling approach.

**Reproducibility:**

3: Could reproduce the results with some difficulty. The settings of parameters are underspecified or subjectively determined; the training/evaluation data are not widely available.

**Reviewer Confidence:**

5: Positive that my evaluation is correct. I read the paper very carefully and I am very familiar with related work.

**Typos Grammar Style And Presentation Improvements:**

- L52: object state(running,
- L576: examples.To

---

> ### Author Rebuttal · Authors · 2023-08-29
>
> Thank you for your constructive comments and we have carefully considered the important issues you raised.
>
> **(a)	Although SGEPG is model-agnostic, this paper conduct experiments using a specific model, VLTVG, to validate the proposed method. The additional experiments using the other VLM would be helpful to emphasize the effectiveness of the scene graph-enhanced pseudo-labeling approach.**
>
> **A:** We appreciate your suggestion for a broader exploration of model compatibility. To underline the efficacy of our scene graph-enhanced pseudo-labeling approach, we have conducted additional experiments using the QRNet [1], in addition to VLTVG. Specifically, we present the results of training QRNet using pseudo-labels generated by our SGEPG method ("SGEPG-QRNet") and the Pseudo-Q approach ("Pseudo-Q-QRNet"). These experiments were conducted on the RefCOCO and RefCOCO+ datasets.
>
> However, it's important to note that due to computational resource limitations and time constraints, we were unable to include the DDQR in our approach when working with QRNet, that further improves performance when using VLTVG. Additionally, we made slight adjustments to the QRNet model's parameter settings during training, including reducing the image size from 640 to 384 pixels and the number of training epochs from 160 to 20. Despite these modifications, our results still offer valuable insights, clearly demonstrating the effectiveness of our SGEPG method. The achieved performance improvements over Pseudo-Q-QRNet underscore the impact of incorporating scene graphs in the pseudo-label generation process.
>
> |                |       | RefCOCO |       |       | RefCOCO+ |       |
> | -------------- | ----- | ------- | ------- | ------- | -------- | ----- |
> |                | val   | testA   | testB   | val   | testA    | testB |
> | Pseudo-Q-QRNet | 56.09 | 59.96 | 54.62   | 37.08 | 44.78    | 32.82 |
> | SGEPG-QRNet    | 62.62 | 62.61 | 60.38   | 44.39 | 47.04    | 40.00 |
>
> *[1] Ye J, Tian J, Yan M, et al. Shifting more attention to visual backbone: Query-modulated refinement networks for end-to-end visual grounding[C]//Proceedings of the IEEE/CVF Conference on Computer Vision and Pattern Recognition. 2022: 15502-15512.*
>
> **(b)	A preliminary scene graph is complemented with the additional information from the off-the-shelf object detector and attribute classifier and spatial information of the bounding box of each object. Have you conducted experiments using a preliminary scene graph, not utilizing a core scene graph? This ablation study would be helpful for readers to understand the impact of the quality of generated scene graphs in the pseudo-query generation process.**
>
> **A:** We appreciate your suggestion and have already conducted the experiments as suggested to assess the impact of using a preliminary scene graph on the pseudo-query generation process. We performed the experiments on both the RefCOCO and RefCOCO+ datasets. In these experiments, we generated pseudo-labels directly using the preliminary scene graph without any further processing.
> Notably, we observed that training the model solely with preliminary scene graphs, which are noisy and unprocessed, led to poor performance. This finding underscores the importance of accurate and refined core scene graphs in our approach. These core scene graphs, complemented with additional information from off-the-shelf object detectors and attribute classifiers, play a crucial role in enhancing the quality of generated pseudo-labels and ultimately improving the performance of the visual-language model.
>
> |                            |       | RefCOCO |       |       | RefCOCO+ |       |
> | -------------------------- | ----- | ------- | ----- | ----- | -------- | ----- |
> |                            | val   | testA   | testB | val   | testA    | testB |
> | preliminary scene graph    | 36.23 | 45.31   | 25.32 | 37.21 | 46.11    | 26.55 |
> | Core scene graph （SGEPG） | 69.61 | 72.34   | 65.67 | 55.21 | 60.67    | 46.76 |
>
> **(c)	This paper adopts GPT-3.5 to make pseudo queries to be diverse and fluent. An additional qualitative analysis of the rewriter would be needed to validate the effectiveness of the rewriting strategy.**
>
> **A:** We appreciate your interest in understanding the impact of linguistic expression rewriting using the GPT-3.5 language model in our work and the effectiveness of the rewriting strategy. However, we would like to clarify a few points in this regard:
>
> 1. First of all, it's important to emphasize that our primary focus is on the utilization of scene graphs to generate high-quality pseudo-labels for ReC. As can be observed from Table 2, while DDQR is a part of our framework and does contribute to further enhancing the model's query diversity and fluency, its impact on the overall model performance is not as substantial as the Scene Graph (SG) and Ambiguity Resolver Strategy (ARS).
> 2. To provide some insights, we conducted an additional experiment to evaluate the semantic consistency of the rephrased sentences generated by DDQR. Among a randomly selected subset of 100 pseudo-label sentences subjected to DDQR rewriting, each sentence was rewritten three times, resulting in 300 rewritten pseudo-labels. Remarkably, 280 of these rewritten statements maintained the same semantic meaning as the original sentences. Additionally, during the rewriting process, out of the 100 original sentences, 87 instances generated multiple rewritten sentences that were consistent with the original sentences.
> 3.  Additionally, there have been studies analyzing the diversity and consistency of rewrites generated by GPT-3.5, as referenced in [1][2]. While our focus was not solely on DDQR, these analyses further support the effectiveness and reliability of the rewriting process in our approach.
>
> *[1]: Jang M, Lukasiewicz T. Consistency analysis of chatgpt[J]. arXiv preprint arXiv:2303.06273, 2023.*
>
> *[2] Feng Y, Qiang J, Li Y, et al. Sentence simplification via large language models[J]. arXiv preprint arXiv:2302.11957, 2023.*

---

### Official Review · Reviewer_3Jqz · 2023-08-10

**Soundness:** 3

**Excitement:**

3: Ambivalent: It has merits (e.g., it reports state-of-the-art results, the idea is nice), but there are key weaknesses (e.g., it describes incremental work), and it can significantly benefit from another round of revision. However, I won't object to accepting it if my co-reviewers champion it.

**Paper Topic And Main Contributions:**

This paper tackles label-efficient unsupervised referring expression comprehension (ReC). Specifically, it presents a scene-graph-based approach to generate high-quality pairs of image regions and linguistic expressions, facilitating joint representation learning at the region level. Initially, the image is embedded into a scene graph using heuristic rules. To tackle the ambiguity inherent in rule-based graph representations of diverse regions, a method for generating region queries is proposed to resolve ambiguity. Additionally, a rewriting module based on GPT-3.5 is employed to enhance linguistic expression diversity. The proposed approach demonstrates clear superiority over previous unsupervised methods.

**Reasons To Accept:**

1. The proposed methods demonstrate impressive performance on benchmarks for unsupervised ReC.
2. Extensive experiments confirm the effectiveness of the proposed model components.
3. The paper is easily understandable and straightforward to follow.

**Reasons To Reject:**

1. This paper presents a highly effective engineering method for ReC. However, it should be noted that the proposed framework incorporates some combinatorial and heuristic aspects. In particular, the Non-Ambiguous Query Generation procedure relies on a sophisticated filtering template. It would be helpful if the author could clarify the impact of these heuristic components.
2. Since the linguistic expression rewriting utilizes the powerful GPT-3.5 language model, it would be interesting to understand the extent of randomness and deviation that may arise from the influence of GPT-3.5. Is there any studies or analyses on this aspect?

**Reproducibility:**

4: Could mostly reproduce the results, but there may be some variation because of sample variance or minor variations in their interpretation of the protocol or method.

**Reviewer Confidence:**

4: Quite sure. I tried to check the important points carefully. It's unlikely, though conceivable, that I missed something that should affect my ratings.

**Typos Grammar Style And Presentation Improvements:**

The pipeline figure (Fig. 2) could benefit from further improvement. For instance, the notation 'F' used in the figure is unclear and could be made more explicit. Additionally, the image layout could be further adjusted to enhance clarity and comprehension.

---

> ### Author Rebuttal · Authors · 2023-08-29
>
> Thank you for your constructive comments and we have carefully considered the important issues you raised.
>
> **(a)	This paper presents a highly effective engineering method for ReC. However, it should be noted that the proposed framework incorporates some combinatorial and heuristic aspects. In particular, the Non-Ambiguous Query Generation procedure relies on a sophisticated filtering template. It would be helpful if the author could clarify the impact of these heuristic components.**
>
> **A:** We appreciate your insightful comments regarding the combinatorial and heuristic aspects of our proposed framework. Indeed, the proposed framework involves some filtering templates, but it's worth clarifying that the templates used in the Non-Ambiguous Query Generation are straightforward, as detailed in Appendix B. Our approach doesn't intricately design complex templates; rather, it populates available subgraph information into basic fillable templates. For instance, a subgraph obtained after applying Ambiguity Resolver Strategy (ARS), like "man-wearing-shirt," is populated into a template like "[sub] [rel] [obj]". However, templates such as "[attr] [sub]" or "[sub] [rel1] [obj1] [rel2] [obj2]" can not be suitable for filling with the information from this subgraph, as it might not completely align with the content of these templates (e.g., missing attribute or additional relationship information).
>
> The ARS, described in Section 3.2 of the paper, serves as the tool to filter subgraphs. The guiding principles of ARS, along with illustrative examples, can be found in Lines 374 to 388 of the paper. We have also conducted ablation experiments in Section 4.5 to demonstrate the effectiveness of ARS, and the results are presented in Table 2.
>
> **(b)	Since the linguistic expression rewriting utilizes the powerful GPT-3.5 language model, it would be interesting to understand the extent of randomness and deviation that may arise from the influence of GPT-3.5. Is there any studies or analyses on this aspect?**
>
> **A:** We appreciate your interest in understanding the impact of linguistic expression rewriting using the GPT-3.5 language model in our work. However, we would like to clarify a few points in this regard:
> 1. First of all, it's important to emphasize that our primary focus is on the utilization of scene graphs to generate high-quality pseudo-labels for ReC. As can be observed from Table 2, while  Diversity Driven Query Rewriter (DDQR) is a part of our framework and does contribute to further enhancing the model's query diversity and fluency, its impact on the overall model performance is not as substantial as the Scene Graph (SG) and Ambiguity Resolver Strategy (ARS).
> 2. To provide some insights, we conducted an additional experiment to evaluate the semantic consistency of the rephrased sentences generated by DDQR. Among a randomly selected subset of 100 pseudo-label sentences subjected to DDQR rewriting, each sentence was rewritten three times, resulting in 300 rewritten pseudo-labels. Remarkably, 280 of these rewritten statements maintained the same semantic meaning as the original sentences. Additionally, during the rewriting process, out of the 100 original sentences, 87 instances generated multiple rewritten sentences that were consistent with the original sentences.
> 3. Additionally, there have been studies analyzing the diversity and consistency of rewrites generated by GPT-3.5, as referenced in [1][2]. While our focus was not solely on DDQR, these analyses further support the effectiveness and reliability of the rewriting process in our approach.
>
> *[1]: Jang M, Lukasiewicz T. Consistency analysis of chatgpt[J]. arXiv preprint arXiv:2303.06273, 2023.*
>
> *[2] Feng Y, Qiang J, Li Y, et al. Sentence simplification via large language models[J]. arXiv preprint arXiv:2302.11957, 2023.*

---

### Meta-Review · Area_Chair_Z3k5 · 2023-09-19

**Recommendation:** 3

**Metareview:**

The paper introduces an unsupervised technique for generating training data for Referring Expression Comprehension (ReC). The proposed method initiates by constructing a core scene graph, subsequently filters out ambiguous queries, and lastly rewrites the query using GPT-3.5 to enhance the quality of the pseudo labels.

When tested on standard benchmarks, this data generation method has shown efficacy across various academic ReC datasets. However, questions about its generalizability have emerged. Notably, the most significant improvement is evident on RefCOCO, which contains spatial terms, making scene graphs apt for capturing principal relations between objects. This might not necessarily translate to datasets outside this specific domain. Additionally, there are reservations about the template-based filtering step's adaptability to diverse distributions and the robustness of its underlying heuristics.

---

### Decision · Program_Chairs · 2023-10-07

**Decision:**

Accept-Findings

**Comment:**

The paper introduces an unsupervised technique for generating training data for Referring Expression Comprehension (ReC). The proposed method initiates by constructing a core scene graph, subsequently filters out ambiguous queries, and lastly rewrites the query using GPT-3.5 to enhance the quality of the pseudo labels.

When tested on standard benchmarks, this data generation method has shown efficacy across various academic ReC datasets. However, questions about its generalizability have emerged. Notably, the most significant improvement is evident on RefCOCO, which contains spatial terms, making scene graphs apt for capturing principal relations between objects. This might not necessarily translate to datasets outside this specific domain. Additionally, there are reservations about the template-based filtering step's adaptability to diverse distributions and the robustness of its underlying heuristics.